# A Bayesian approach for fitting and comparing demographic growth models of radiocarbon dates: A case study on the Jomon-Yayoi transition in Kyushu (Japan)

Enrico R. Crema[1]*, Shinya Shoda[2,3]

**1** Department of Archaeology, University of Cambridge, Cambridge, United Kingdom, **2** BioArCh, University of York, Wentworth Way, Heslington, York, United Kingdom, **3** Nara National Research Institute for Cultural Properties, Nara, Japan

\* erc62@cam.ac.uk

## Abstract

Large sets of radiocarbon dates are increasingly used as proxies for inferring past population dynamics and the last few years, in particular, saw an increase in the development of new statistical techniques to overcome some of the key challenges imposed by this kind of data. These include: 1) null hypothesis significance testing approaches based on Monte-Carlo simulations or mark permutations; 2) non-parametric Bayesian modelling approaches, and 3) the use of more traditional techniques such as correlation, regression, and AIC-based model comparison directly on the summed probability distribution of radiocarbon dates (SPD). While the range of opportunities offered by these solutions is unquestionably appealing, they often do not consider the uncertainty and the biases arising from calibration effects or sampling error. Here we introduce a novel Bayesian approach and *nimbleCarbon*, an R package that offers model fitting and comparison for population growth models based on the temporal frequency data of radiocarbon dates. We evaluate the robustness of the proposed approach on a range of simulated scenarios and illustrate its application on a case study focused on the demographic impact of the introduction of wet-rice farming in prehistoric Japan during the 1st millennium BCE.

## 1. Introduction

Radiocarbon dates have been used as a proxy for investigating past human population dynamics for over three decades, but the last few years saw an exceptional increase in the number of studies thanks to the broader availability and accessibility of large collection of [14]C dates (e.g. [1–6]) as well as dedicated software packages enabling the use of an array of new statistical approaches [7–12]. The opportunities offered by this growth are unquestionable [13]; we are no longer constrained by the arbitrary limits of regional relative chronologies, we can test the supposed role of climate change on demographic processes [14–21], and pursue comparative studies at continental [22–26] or even global scales [27].

(https://github.com/ercrema/bayesian_spd) and on the following archive http://doi.org/10.5281/zenodo.4738545

**Funding:** ERC was funded by: 1) European Research Council Starting Grant for the project 'Demography, Cultural Change, and the Diffusion of Rice and Millets during the Jōmon-Yayoi transition in prehistoric Japan (ENCOUNTER)', Project N. 801953, PI: Enrico Crema. URL: https://erc.europa.eu/ 2) Philip Leverhulme Prize (Grant Number: PLP-2019-304), URL: https://www.leverhulme.ac.uk/ Neither of the sponsors played a role in the study design, data collection and analysis, decision to publish, or preparation of the manuscript

**Competing interests:** The authors have declared that no competing interests exist.

However, this endeavour comes with its own set of inferential challenges that have raised a substantial amount of criticisms over the past few years [28–33]. On the one hand, the issue concerns whether fluctuations in the density of radiocarbon dates truly reflect corresponding changes in human populations [34]. The problem arises because the so-called 'dates as data' approach [35] is pragmatically vague on the definition of what the events associated with each $^{14}$C sample constitutes (cf. [30] for an early criticism on this issue). This assumption inevitably leads to the question of what exactly is being counted, with both interpretative and methodological consequences (see for example the implications of 'binning' samples recovered from the same context to control for inter-site variation in sampling intensity [8]). On the other hand, the analyses of radiocarbon dates have also specific statistical challenges. Some, such as the impact of sampling error, are not unique to this kind of data and although occasionally ignored, can be tackled straightforwardly. Others, such as the systematic and idiosyncratic errors introduced by the calibration process, are more complex and require the development of bespoke solutions.

The most commonly adopted approach for examining the time-frequency of radiocarbon dates consists of aggregating the calibrated probability distribution of all samples and generating what is commonly referred to as *summed probability distribution of radiocarbon dates* (hereafter SPD). While this and similar approaches such as composite kernel density estimates [36] do provide a quick way to visually assess the observed changes in the frequency of radiocarbon dates, they ignore the issue of sampling error and calibration effects. As a result, visual inspections of SPDs can be misleading and should never be used as the sole mode of assessment of the available evidence. Some attempts have also been made to address the problem of sampling error (e.g. via bootstrapping, [9, 21, 35]) or calibration effect (e.g. via moving window, [14, 22]), but none can satisfactorily solve all the problems simultaneously.

More formal statistical treatment of radiocarbon dates are clearly necessary for proper inference, and while there have been a fairly large number of solutions, time-frequency of radiocarbon dates have been generally analysed in three ways: 1) using Monte-Carlo simulation or random permutation techniques to carry out null hypothesis significance testing (NHST) [22, 37, 38]; 2) fitting Bayesian non-parametric models to reconstruct the underlying 'shape' of the probability distribution of the $^{14}$C dates [7, 11]; and 3) employing conventional statistical methods (e.g. correlation and regression analyses, information-criterion based model comparison, etc.) directly on SPDs (for example [14, 15, 19, 22, 39–43]). The first two approaches can be somewhat limited in their scope but were designed to specifically address issues pertaining to sampling error and calibration effects directly. The third group of applications offer a substantially wider range of applications and, in many cases, address more important sets of research questions as they often embrace a model-based inference that could potentially draw and expand from the rich literature of theoretical population models developed in ecology (see for example [15]). However, from a statistical viewpoint, these approaches commonly hinge on the assumption that the vector of summed probability values per calendar year can be directly and reliably used for statistical inference. As others have already recently noted [10, 31] this is unwarranted, as it simultaneously ignores sampling error and calibration effects.

This paper presents a new approach that addresses these concerns by considering the problem to be similar to the objectives and principles of Bayesian phase modelling [44–46]. More specifically, we introduce a flexible family of bounded growth models with discrete time units that can be fitted via Markov Chain Monte-Carlo (MCMC) and compared using the Widely Applicable Information Criterion (WAIC, [47]). To determine the robustness and the potential of the proposed approach, we tested our new method on a series of simulated datasets and a case study. The former was designed to determine whether our Bayesian inferential

framework can reliably recover parameters of interest under controlled realistic scenarios. In the latter, we employ our new method to explore the demographic consequences of the introduction of irrigated rice farming in the island of Kyushu (south-west Japan) during the 1st millennium BCE, aiming to determine whether and when we can observe a shift in population growth rate, and to measure the magnitude of the population boom promoted by the new subsistence strategy.

## 1.1 Fitting and comparing demographic growth models using radiocarbon dates

The core statistical challenge of SPD analyses is that the combination of sampling error and the systematic measurement error introduced in the calibration process hinders our ability to visually discern genuine fluctuation in the density of radiocarbon dates from those associated with these and other factors [31, 48, 49]. The problem is further exacerbated by the fact that, in contrast to typical distributions of archaeological samples, the true shape of the underlying population distribution is unknown and reliable expectations cannot be presumed on the basis of general principles such as the central limit theorem. As such, we have no intuition to aid our visual assessments — any peak or trough in the observed SPD can be plausibly interpreted as evidence of a fluctuation in the frequency of radiocarbon dates and naively support the claim for the existence of a population rise or fall. The problem has been widely discussed in the literature [48–50], and has shown that in extreme cases, one can have synchronic peaks in SPDs generated from samples from different continents purely as the consequence of the calibration process (see Fig 2 in [8]).

A simple, visual assessment of SPD is thus unwarranted, and so is the direct use of SPD for correlation and regression analyses. Tests on simulated datasets [31] have shown, in fact, that simple regression analyses of time against corresponding summed probability values fail to recover true slope values. Furthermore, because such an approach effectively implies as many observation points as the number of years in the window of analyses, the actual number of radiocarbon samples is not formally taken into consideration. The ultimate consequence is that statistical measures such as standard errors of fitted parameters, P-values, and likelihood estimates (and derived statistics such as AIC) are all biased [10].

A number of approaches have been introduced in the last few years to effectively tackle these problems. One of the most commonly adopted solutions involves the use of Monte-Carlo simulations for null hypothesis significance testing (NHST), whereby an ensemble of SPDs generated from a particular growth model is compared against the observed data [22, 37]. The approach consists of sampling $n$ calendar dates from the hypothesised growth model, back-calibrate each sample in $^{14}C$ age, calibrate back in calendar time, and aggregate to generate a putative SPD under the null hypothesis. The process is repeated multiple times, so that an envelope of SPDs can be generated and compared to the observed data. The approach has been subsequently extended to include post-hoc tests [51], mark-permutation tests for comparing different sample sets [38] including spatial locations [24], as well as different algorithms for generating samples from the null model [8, 20]. The shared principle of all these approaches is to treat SPDs as a form of test statistic rather than a direct proxy of demographic fluctuations.

While the one-sample, MC simulation-based NHST is a convenient way to quickly assess the presence of significant episodes of anomalies against a putative growth process, it is limited by three issues. Firstly, the definition of the null model is central, and as such, detections and interpretations of episodes of population 'boom' or 'busts' can be severely conditioned by its choice. Secondly, the null hypothesis is typically based on a particular growth model (e.g.

exponential, logistic, etc.), which parameter values are often derived by fitting the observed SPD via regression analysis. Aside from issues of circularity, such a procedure implies that the test is carried out on one particular case (i.e. a specific parameter combination) of a given growth model and as such, its rejection (or 'acceptance') does not necessarily extend to other parameter combinations of the same type of growth model. Furthermore, the potential biases in regression-based fitting of SPDs [31] imply that there is no guarantee that the test is carried out against the best parameter combination for a particular model (it is worth noting, however, that the approach does not necessarily have to be based on parameter combinations derived by regression-based fitting, and can easily be based on theoretical expectations or independent proxies, see for example [52]). Thirdly, as with any other NHST approaches, there is a limited inferential gain by simply rejecting (or failing to reject) particular null hypotheses. Clearly, no time-frequencies of radiocarbon dates follow exactly the expected pattern from a given growth model, and as such, when sufficiently large sample sizes are available, obtaining significant P-values becomes trivial. This problem also applies to mark-permutation tests and their spatial variants [24, 38], although these are not affected by how the null hypothesis is being defined.

A second group of approaches involves the direct reconstruction of the shape of the underlying time-frequency distribution through non-parametric Bayesian models. Examples of this category include Bayesian KDE [7] and Gaussian mixture approaches [11] (see also *Bchron-Density* function in the *Bchron* R package [53]). While these examples differ in key details, they fundamentally share the same objective of inferring the shape of the underlying population distribution whilst acknowledging the uncertainty associated with sampling and calibration. The main advantage of these solutions is that they do not require the formulation of explicit hypotheses and models, and as such, their outputs can replace the visual assessment of SPDs. While some parameters, such as growth rates for user-defined intervals, can be extracted from these models [11], these techniques are limited if one wishes to examine the impact of covariates [15] or compare alternative growth models derived from explicit theoretical principles [40, 43].

Two works have recently tackled these problems by offering a novel perspective that substantially differs from the NHST and the non-parametric approaches discussed above. The first [12] treats the time-frequency of radiocarbon dates as a special case of a one-dimensional temporal point process. The core concept of this method consists of a two-stage analyses where calendar dates from observed calibrated radiocarbon dates are first sampled to generate an ensemble of discrete count time-series that are then iteratively fitted using an inhomogeneous Poisson model. The solution offers the flexibility of generalised linear models whilst accounting for the uncertainty associated with radiocarbon dates. As a result, one can virtually fit any growth model, including those with external covariates. However, because at its core this approach is estimating counts per time-unit, it does not discern evidence of absence from absence of evidence (i.e. the sample size, in this case, is the number of time-units rather than the number of radiocarbon dates) and further studies are required to establish its robustness under small sample regimes.

Timpson and colleagues [10] have instead tackled the problem by using a maximum likelihood approach. Their key insight was to solve the mathematically intractable problem of calculating the likelihood function of growth models by treating time as discrete rather than continuous. By modelling the distribution of observed dates as a probability mass rather than a probability density, they managed to compute likelihoods from the probability distribution of calibrated dates. This enabled them to carry out model comparison via BIC and obtain estimates of the uncertainty of the fitted parameters via MCMC. There are further advantages in using a likelihood-based approach under this framework. One example explored by the authors is the possibility to employ solutions such as continuous piecewise linear (CPL)

models, which marry the flexibility of non-parametric approaches whilst returning interpretable and comparable parameters. Here we introduce a novel approach that combines this key insight of using discrete probability distributions with the long-established Bayesian framework for modelling archaeological phases [44, 46].

## 1.2 Bayesian analysis of bounded growth models

Bayesian analyses of archaeological phases treat observed radiocarbon dates as random and independent samples from a probability distribution, most commonly a uniform probability distribution with a start date $a$ and an end date $b$. Thus, typically, the objective is to infer the values of $a$ and $b$ given the observed data, constraints, and priors [44]. The approach can be extended to a variety of probability distributions, and dedicated software packages such as *OxCal* [54] can now offer a wide range of options, including exotic distributions such as the trapezoid [55].

The core aims of these Bayesian analyses are not that different from the objectives shared by those attempting to infer prehistoric population change from $^{14}$C samples. Dates are still generally treated as random and independent samples from a population that is changing its size over time and can be conceptualised as a probability distribution. The key assumption is that the probability that a radiocarbon sample is coming from time $t$ is proportional to the relative size of the human population at $t$. In contrast to phase models, where the primary goal is typically to estimate the boundaries of a distribution, here, we focus on estimating its shape. However, the range of possible shapes offered by probability distributions where the likelihood can be mathematically computed is limited, and as such, there is little applicability offered by currently available software packages.

The problem can be solved by treating time as discrete [10] and employing probability mass functions instead of probability density functions. Such a solution effectively provides a framework for fitting any theoretical growth model as long as its associated parameters can generate a vector of probability values for each calendar year $t$. More formally, we can define a *bounded growth model* as a generalised Bernoulli distribution where the probability of sampling a calendar date $t$ is given by the associated probability $p_t$, which is part of a vector of probabilities $p_{t = a-0}, p_{t = a-1}, p_{t = a-2}, \ldots, p_{t = a-T}$, where $a$ and $b$ are the start date in BP, $T$ is equal to $a-b$, and $p_{t>a} = p_{t<b} = 0$. The specific probabilities assigned to each calendar year between $a$ and $b$ are determined by the parameters of each specific growth model. In practice, any numerical model of population growth that is able to generate a vector of population sizes $N_t$ for each time $t$, can be translated into such bounded growth model. For example, a bounded exponential growth model can be formally defined as follows:

$$p_{t=a-i} = \frac{(1 + r)^i}{\sum_{i=0}^{a-b} (1 + r)^i} \tag{1}$$

The numerator, in this case, is a simplified version of an exponential growth model $N_0(1 + r)^i$, where $N_0 = 1$, and the denominator ensures that the total sum of the sequence $p_{t = a-0}, p_{t = a-1}, p_{t = a-2}, \ldots, p_{t = a-T}$ is equal to 1. Because we are just interested in the relative shape of the distribution, the constant $N_0$ becomes obsolete, and effectively the entire model can be described with just the three parameters $a$, $b$, and $r$. If we account for calibration and the Gaussian measurement error associated with each observed $^{14}$C age, we can define our full

hierarchical Bayesian model as follows:

$$\theta_j \sim BoundedExponentialGrowth(a, b, r) \tag{2}$$

$$X_j \sim Normal(\mu(\theta_j), \sigma_j) \tag{3}$$

where $\theta_j$ is the true calendar date of the sample $j$, $\mu(\theta_j)$ is its corresponding $^{14}$C age, $\sigma_j$ is the root of the sum of the squares of the sample $^{14}$C age error and the corresponding error in the calibration curve, and $X_j$ is the observed $^{14}$C age of the sample. Eq (2) can be replaced by any growth models satisfying the criteria described above, whilst Eq (3) effectively models the measurement error associated with the $^{14}$C sample. Posterior samples of $\theta_j$ and the parameters associated with the given growth model (in this case $a$, $b$, and $r$) can be obtained using Markov Chain Monte Carlo (MCMC) methods.

In addition to the use of discrete rather than continuous time, in most practical applications we would need to consider another key difference between phase and growth models. Typically, when modelling an archaeological phase, the selection of samples is defined by explicit constraints such as their location (e.g. a specific stratigraphic unit) or some other shared characteristics (e.g. association to a particular material culture type). In the case of growth models, such definitions are not necessarily always clear. For example, one might be interested in modelling a population trajectory only within a specific time interval. In these circumstances, the parameters $a$ and $b$ can be treated as user-defined constants as long as the sample selection conforms with those boundary conditions (i.e. they have a true calendar date between $a$ and $b$). In practice, however, this process is not straightforward as measurement errors and the information loss after calibration do not allow a clear assignment of all dates within and without the temporal window defined by $a$ and $b$. A simple solution, in this case, consist of computing the cumulative probability of each calibrated date within the window of analysis and consider only samples above some threshold probability value (see also [10]).

## 1.3 Case study: The Jomon-Yayoi transition in Kyushu, Japan

The 1st millennium BCE is a defining period within the Japanese islands triggered by a major demic and cultural diffusion event that led to profound and long-lasting social, economic, cultural, and biological changes [56–58]. Archaeologically this moment is marked by the transition from the so-called *Jomon* period (ca 14,000 –early 1$^{st}$ millennium BCE), characterised predominantly (but not exclusively) by a hunting and gathering economy, to the *Yayoi* period (early 1st millennium BCE–ca 250 CE), typically associated with the introduction of wet-rice farming brought in by migrant communities from the Korean peninsula in northern Kyushu during the first few centuries of the 1st millennium BCE. From a genetic standpoint, the subsequent admixture between the migrant and the incumbent groups is recognised as the origin of the modern Japanese population [59, 60].

The chronological demarcation between the Jomon and Yayoi periods is, however, not trivial. Firstly, the diffusion across Japan of key traits such as rice cultivation occurred over several centuries. Furthermore, in some regions, archaeological evidence suggests only a temporary adoption (followed by a reversion to the previous subsistence economy) or significant delays until historical periods [61]. Thus, a 'start date' of the Yayoi period would vary geographically, with a general gradient moving from west to east following the dispersal of the Yayoi package [57]. A second issue spawns from the complexity of how the available radiocarbon evidence should be interpreted in relation to the archaeological record, particularly in the northern part of the island of Kyushu, which is regarded as the entry point of the migrant population and the first area where the Jomon-Yayoi transition took place. While traditionally the start of the

Yayoi period was dated to 300 BCE (or 500 BCE; see [62, 63] for a review), in 2003, a team of the National Museum of Japanese History carried out a large-scale AMS dating project that led to a controversial shift in chronology that pushed back the start of the Yayoi period from ca. 500 BCE to the start of the millennium [64]. This has led to a major debate within the archaeological community, and while there is now a substantial agreement for an earlier start date compared to the traditional chronology, scholars have taken different viewpoints and the beginning of the Yayoi period is currently anywhere between 1,000 BCE to 600 BCE.

The difference in these chronologies can have profound interpretative implications, particularly when examining the relative contribution of demic versus cultural diffusion processes on the spread of continental traits or evaluating the demographic impact of irrigated rice farming. While the presence of a population 'boom' following the introduction of wet-rice farming is evident from both archaeological and genetic lines of evidence [65, 66], there is substantial uncertainty regarding its magnitude and timing. These factors, however, play a central role when attempting to estimate the actual size of the initial migrant population or the frequencies of migratory events. In particular, several demographic simulation models have been developed to explore putative demic and cultural dynamics in relation to specific assumptions and variables drawn from the available evidence [67–71]. These are typically based on estimates of population growth rates and the time span of the admixture between incumbent and migrant populations. It is thus not a surprise that chronological revisions of the Yayoi start date led to different estimates of these variables, and in some cases, prompted a re-evaluation of previous works on Yayoi demography [70, 72].

Population growth estimates of the Yayoi period for the island of Kyushu (S1 Fig) are limited to a few number of studies based on common archaeological proxies such as count frequencies of sites, residential units, and burials. Shuzo Koyama's widely cited study [65] utilised the number of archaeological sites across the Jomon and Yayoi periods in conjunction with historical census data to estimate absolute population sizes across different regions in Japan while taking into account issues such as sampling fraction and settlement size variation. His estimated annual growth rate between the Final Jomon period and the Yayoi period in Kyushu is 0.26% [73]. Other studies have focused exclusively on relative population sizes and examined the temporal frequencies of archaeological features per ceramic phase. Nakahashi [74] use the frequencies of jar-burials and obtained an estimate larger than >1% for the Middle Yayoi period. More recently, Kataoka and Iizuka [75] fitted exponential and logistic growth models to the time-frequency of residential features, obtaining various estimates as a function of different assumptions on the duration of ceramic phases. A follow-up study by Fujio [72] on the same dataset yielded an estimate of 0.8% with an exponential fit over an interval of 400 years.

Growth rate estimates derived from the time-frequency of radiocarbon dates are currently not available for the Jomon-Yayoi transition, despite their use in other parts of the world that have already prompted cross-cultural discussions on the demographic impact of early farming [25, 76]. While some have highlighted potential issues in interpreting growth estimates derived from the time-frequency of radiocarbon dates (particularly when compared to the shorter time-scale of ethnographic studies [77]), the substantial discrepancies on the current range of estimates derived from other proxies in the island of Kyushu is problematic. The issue is further exacerbated by the fact that most count data are not associated with absolute dates, and as such time-frequencies are strongly dependent on assumptions of contemporaneity, absolute estimates of relative chronological sequences, and the treatment of different forms of uncertainties [52] as well as intrinsic biases derived by how the time is subdivided into archaeological periods [78].

Here we used the Bayesian approach described in section 1.2 to offer an alternative means for estimating the magnitude of the Yayoi population boom that is independent of

archaeological discussions and updates of Yayoi chronology. More specifically, we focus on the changing frequency of radiocarbon dates over the temporal window between 3400 and 1850 cal BP (1451 BCE to 100 CE), corresponding approximately to the time interval between the Final Jomon and the Middle Yayoi periods. We expect that during this interval, there was a significant shift in the population growth rate following the introduction of wet-rice farming, and hence fit two different types of growth models (see section 3) that account for such a change. Thus, our objective is to determine whether there is evidence in support of a change in the growth rate during the examined time interval and, if so, to estimate when this change occurred and what was the population growth rate before and after this point in time.

## 2. Materials and methods

### 2.1 Simulation data

A total of four sets of experiments (see Table 1) were performed in order to determine the robustness of the proposed method under different scenarios. In all cases, a sample of calendar dates was generated from a particular bounded growth model and converted into $^{14}C$ age with a fixed $^{14}C$ error of 20 years.

Experiment 1 was designed to determine whether the method was capable of correctly inferring the growth rate $r$, using a sample of 500 dates from three different bounded exponential growth models with $a = 6,000$ cal BP, $b = 4,000$ cal BP and $r$ set to 0.001, 0.003, and 0.005. Twenty replicates were generated for each of the three settings of $r$. The objective of experiment 2 was to determine the impact of the calibration effect in the inferential process. A total of 2 x 20 sets were examined, each consisting of a sample of 300 dates drawn from an exponential model with $r = 0.003$, and with two different boundary parameters reflecting time intervals where the calibration curve was characterised by a steep slope ($a = 7,000$ cal BP; $b = 6,400$ cal BP; 20 replicates) or a plateau ($a = 2,800$ cal BP; $b = 2,200$ cal BP; 20 replicates).

**Table 1. Parameter settings and priors for experiments 1–4.**

| No. | Growth Model | Parameters | Parameter Values | Priors | Sample Size |
|---|---|---|---|---|---|
| 1 | Exponential | $r$ | 0.001, 0.003, 0.005 | Exponential; λ = 500 | 500 |
| | | $a$ | 6000 | - | |
| | | $b$ | 4000 | - | |
| 2 | Exponential | $r$ | 0.003 | Exponential; λ = 500 | 300 |
| | | $a$ | 7000, 2800 | - | |
| | | $b$ | 6400, 2200 | - | |
| 3a | Exponential | $r$ | 0.005 | Exponential; λ = 500 | 250, 100, 50 |
| | | $a$ | 6000 | - | |
| | | $b$ | 4000 | - | |
| 3b | Double-Exponential | $r1$ | 0.002 | Normal; μ = 0, σ = 0.1 | 500, 250, 100, 50 |
| | | $r2$ | -0.001 | Normal; μ = 0, σ = 0.1 | |
| | | $c$ | 4500 | Uniform; a = 5999, b = 4001 | |
| | | $a$ | 6000 | - | |
| | | $b$ | 4000 | - | |
| 4 | Double-Exponential | $r1$ | -0.001, -0.002, -0.003 | Normal; μ = 0, σ = 0.1 | 288 |
| | | $r2$ | 0.001, 0.002, 0.003 | Normal; μ = 0, σ = 0.1 | |
| | | $c$ | 2800 | Uniform; a = 5999, b = 4001 | |
| | | $a$ | 3400 | - | |
| | | $b$ | 1850 | - | |

Experiment 3 examined the impact of sample size in the inferential process. In 3a we examined an exponential growth model with parameters $a$ = 6,000 cal BP, $b$ = 4,000 cal BP, and $r$ = 0.005 using sample sizes of 250, 100, and 50 dates, and 20 replicates for each setting. In order to explore the impact of smaller sample sizes on more complex models, an additional experiment (3b) was carried out using a double exponential growth model (see section 3 for definition) with parameters $a$ = 6,000 cal BP, $b$ = 4,000 cal BP, $r_1$ = 0.002, $r_2$ = −0.001, and $c$ = 4,500 cal BP, and sample sizes 500, 250, 100, and 50 with 20 replicates for each setting.

Finally, to establish whether the sample size used in the case study (see below) was sufficient for correctly detecting the timing and the magnitude of a putative population boom in Kyushu, we carried out an experiment (experiment 4) with a double exponential growth model using the following parameters: $a$ = 3,400 cal BP; $b$ = 1,850 cal BP; $c$ = 2,800 cal BP, $r_1$ = {−0.001, −−0.002, −0.003} and $r_2$ = {0.001, 0.002, 0.003}. These settings cover the same chronological window of our case study with an annual percentage growth rate comparable to those observed in previous work (i.e. 0.1 ∼ 0.3%). We sampled 288 dates (the same number of dates in the case study, see section 2.2) for each of the nine parameter combinations of the growth rates $r_1$ and $r_2$ (see S2 Fig).

## 2.2 Empirical data

Radiocarbon dates for the case study have been collated from the National Museum of Japanese History's radiocarbon database [79], using as a search criterion terrestrial samples from Fukuoka, Kagoshima, Miyazaki, Nagasaki, Oita, and Saga prefectures in the island of Kyushu (S1 Fig). No permits were required to access and use the materials which complied with all relevant regulations. The initial downloaded dataset has been pre-processed by excluding samples with 1) no dates; 2) no lab codes; 3) $^{14}$C error above 100 years (to minimise the impact of measurement error); 4) originating from contexts that are not clearly anthropogenic; and 4) with a calibrated cumulative probability between 3400 and 1850 cal BP below 0.5. Marine samples and terrestrial samples from bones were excluded from the analysis due to the lack of reliable reservoir correction measures for the region. The resulting final sample size consisted of 288 radiocarbon dates from 131 unique site locations covering a chronological span corresponding approximately to the interval between the Final Jomon and the Middle Yayoi periods, with the SPD showing an increase in the density of dates during the latter half of the window of analyses (Fig 1).

## 3. Methods

Posterior samples for all experiments and the case study were obtained using a Metropolis-Hastings adaptive random-walk sampler with a univariate normal proposal distribution using the *nimble* R package version 0.10.1 [80, 81]. The simulated data have been analysed by comparing the 'true' parameter values of each setting against the 90% higher posterior density interval (HPD) estimated from the posterior samples, obtained using a single chain with 10,000 iterations and 3,000 burn-in steps and prior distributions as reported in Table 1.

The empirical data set has been analysed using three growth models (Fig 2). First, we examined a simple exponential growth model (model **m1**) as described in section 1.2 (Eq 1; Fig 2, left). The model would be consistent with a scenario where there is no noticeable change in the growth rate over the window of analyses. We then considered a double exponential growth model (model **m2**; Fig 2, centre) where the exponential growth rate $r$ was set to $r_1$ for $t > c$ and equal to $r_2$ for $t \leq c$. In other words, the model assumes two exponential growth rates, with the parameter $c$ defining the change point (in cal BP) when the growth rate shifts from $r_1$ to $r_2$. In this particular context, a support for this model with $r_2 > r_1$ would be consistent with a scenario

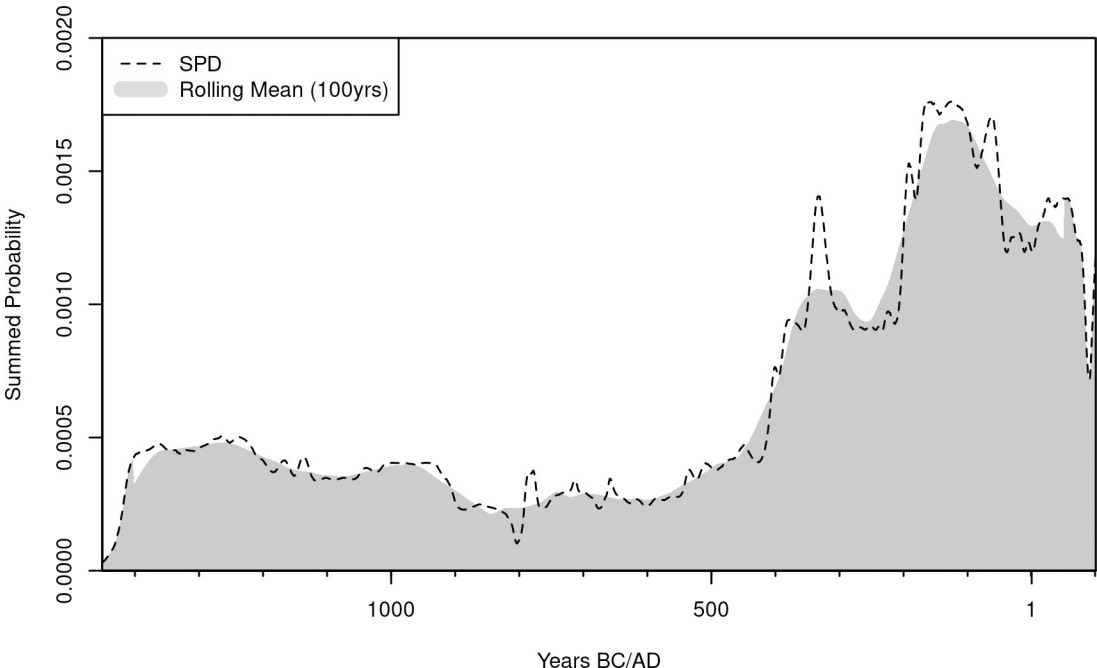

**Fig 1. Observed Summed Probability Distribution (SPD) of radiocarbon dates for the island of Kyushu.**

where the adoption of irrigated rice farming led to an increase in growth rate, with $c$ providing an estimate of when such demographic shift took place in relation to the introduction of the new subsistence technology.

Finally, we examined an exponential-logistic growth model (model **m3**; Fig 2 right) defined as follows:

$$p_{t=a-i} = \frac{h_1 k (1 + r_1)^i + h_2 \left( \frac{1}{1 + \frac{1 - k(1+r_1)^{a-c}}{k(1+r_1)^{a-c}} e^{-r_2 i}} \right)}{\sum_{i=0}^{a-b} h_1 k (1 + r_1)^i + h_2 \left( \frac{1}{1 + \frac{1 - k(1+r_1)^{a-c}}{k(1+r_1)^{a-c}} e^{-r_2 i}} \right)} \tag{4}$$

Where $r_1$ is the growth rate in the exponential part (when $a \geq t > c$), $r_2$ is the intrinsic growth rate of the logistic growth part (when $t \leq c$), $k$ is the proportion of the population relative to the carrying capacity at the time $t = a$, $c$ is the changepoint in calendar BP between the exponential and the logistic growth part, and $h_1$ and $h_2$ are indicator variables, with $h_1 = 1$ and $h_2 = 0$ when $t > c$ and $h_1 = 0$ and $h_2 = 1$ when $t \leq c$. Model **m3** can effectively be considered a particular variant of the two-stage demographic process captured in model **m2**. The key difference, in this case, is that the net growth rate after the change point $c$ can potentially decline as the population approaches carrying capacity.

In all three models, the probability of each $i$-th timestep (corresponding to the calendar year $t$ in BP, i.e. $a-i$) were calculated using *nimbleCarbon* v0.1.0 [82], a dedicated R package developed for this purpose which includes several bounded growth models as well as utility functions for fitting and comparing growth models.

We used weakly informative priors for all three models, which allows the 'data to speak' whilst simultaneously providing some degree of regularisation. Prior predictive checks (S3 Fig) were explored to evaluate the impact of the priors for each model. Results show a wide

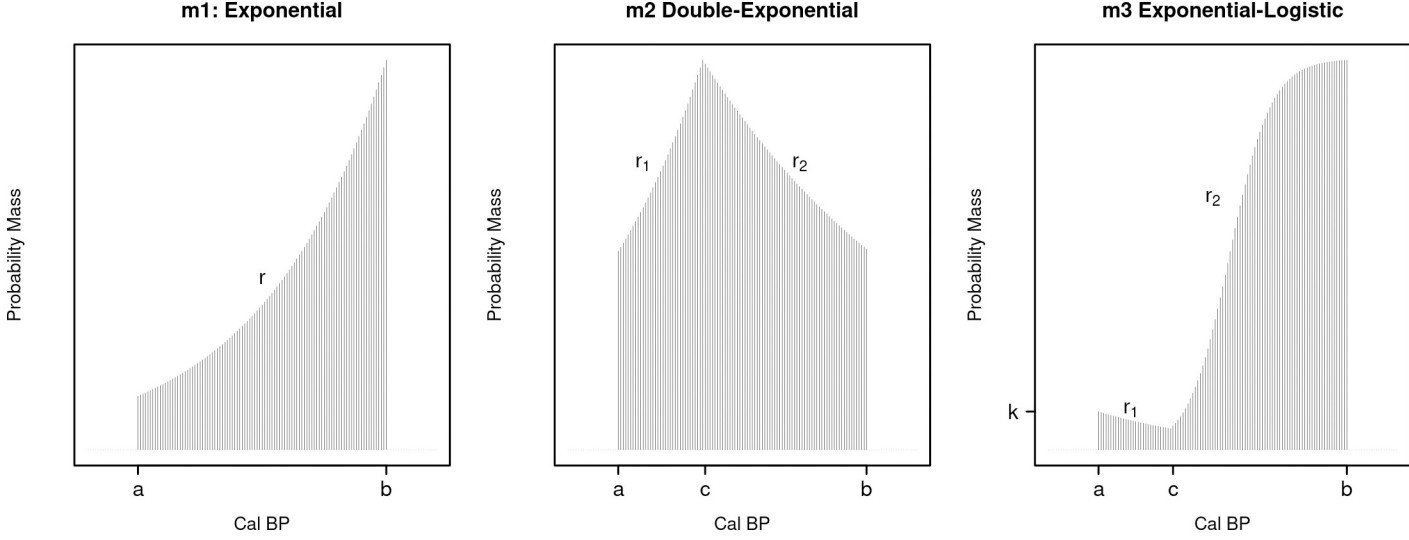

**Fig 2. Models m1, m2, and m3 with associated parameters.**

range of patterns that are however consistent within a plausible range for human population dynamics. For model **m1**, we estimated the parameter *r* using as prior an exponential distribution with $\lambda = 1/0.0004$, with the denominator based on the average growth rate observed across multiple prehistoric populations [76] (see S3 Fig, left, for prior predictive checks). For model **m2** we used the following priors:

$$r_1 \sim Normal(\mu = 0, \sigma = 0.0004) \tag{5}$$

$$r_2 \sim Exponential\left(\lambda = \frac{1}{0.0004}\right) \tag{6}$$

$$c \sim TruncatedNormal(\mu = 2625, \sigma = 200, a = 1850, b = 3400) \tag{7}$$

This ensured that the first exponential phase can capture the possibility of a population decline (as well as increase) and that the change point is less likely to occur at the edge of the time range of analyses (see S3 Fig, centre, for prior predictive checks). The priors for the parameters $r_1$, $r_2$ and $c$ for model **m3** were the same as **m2**, whilst the prior for the initial population size (proportion of carrying capacity) *k* was set as a truncated normal distribution with a mean of 0.1, and standard deviation of 0.1, bounded between 0.0001 and 0.5 (see S3 Fig, right, for prior predictive checks). These settings were loosely based on estimates of the ratio of Jomon and Yayoi population sizes used in other works (for example [69]).

For each model, we ran three chains with 100,000 iterations and 10,000 burn-in steps and thinned the output to obtain a total of 15,000 posterior samples. We examined the convergence and mixing by visually assessing the trace plots and by calculating the effective sample size and the Gelman-Rubin's convergence diagnostic [83] as implemented in the *coda* package in R [84]. The Gelman-Rubin's diagnostic provides a heuristic measure for monitoring the convergence of MCMC samples across different chains and provides the basis for determining whether the number of iterations is sufficient for obtaining reliable posterior samples, with a threshold typically set at $\hat{R} \leq 1.001$. (i.e. convergence is typically reached when the diagnostic is smaller than 1.001).

We carried out posterior predictive checks by visually comparing the observed and an envelope of simulated SPDs. The latter was based on 500 different curves, each obtained by 1) sampling random calendar dates from fitted models based on posterior parameter combinations; 2) back-calibrating each date into $^{14}C$ age; 3) randomly assigning $^{14}C$ errors by resampling values from the observed data; 4) calibrating, and 5) aggregating to generate an SPD. The procedure was carried out using a modified version of the *modelTest* routine used in *rcarbon* [8]. Finally, in order to compare the relative goodness-of-fit among the three growth models, we used the Widely Applicable Information Criterion (WAIC) [47, 85]. The WAIC is a more advanced and general information criterion to AIC as it does not require flat priors nor the posterior to be multivariate Gaussian.

All analyses used the *IntCal20* calibration curve [86]. In order to ensure full reproducibility [87], source data, R scripts, and MCMC outputs are available on a GitHub repository (https://github.com/ercrema/bayesian_spd) and on the following archive http://doi.org/10.5281/zenodo.4738545).

## 4. Results

### 4.1 Simulation experiments

All experiments have shown that the proposed method can correctly infer true model parameters without any discernible directional biases, with the number of cases where the true parameter value was outside the predicted interval consistent within the expectations of a 90% HPD. Results of experiment 1 (Fig 3A) indicate that only six sets out of a total of 60 failed to include the correct value of *r* in their 90% HPD interval. Although a thorough exploration of different regions of the calibration curve was not viable, the results of experiment 2 (Fig 3B) do not suggest any tangible impact caused by the presence of plateaus in the calibration curve, at least when estimating a growth rate for a simple exponential model. Unsurprisingly, smaller sample sizes (experiment 3a, Fig 4) do enlarge the HPD interval, but there is no evidence suggesting an increase in inferential errors, with the true growth rate included in the 90% HPD interval in 18 out of the 20 cases even with a sample size as small as 50 radiocarbon dates. However, results of experiment 3b (Fig 5) show that with more complex models such as a double exponential growth, sample sizes of 50 or even 100 dates are not sufficient for achieving reliable estimates, with large HPD intervals and a slightly higher number of cases with incorrect inference. Given the relatively smaller number of MCMC iterations in these experiment runs, there is a possibility of some improvement in the posterior estimates, although we did not explore this in detail due to the high computational cost. The results of experiment 3b strongly suggest that, with all things being equal, the minimum number of radiocarbon dates required to reliably estimate model parameters is higher for more complex models.

Experiment 4 was designed specifically to determine whether plausible dynamics of a double exponential model can be inferred with the observed sample size and window of analysis. Results (Fig 6) suggests that is the case with the proposed method capable of correctly recovering the correct growth rates and change points for a range of different scenarios and in the majority of the replicates.

### 4.2 Case study

Convergence diagnostics and trace plots (S4 to S6 Figs) indicate adequate mixing of the chains, with $\hat{R} \leq 1.001$ and an effective sample size larger than 10,000 for all key parameters. Table 2 and Fig 7 provides the 90% HPD intervals and the marginal posterior distribution of the key parameters of all models. The posterior distribution of the exponential model (**m1**) yielded a

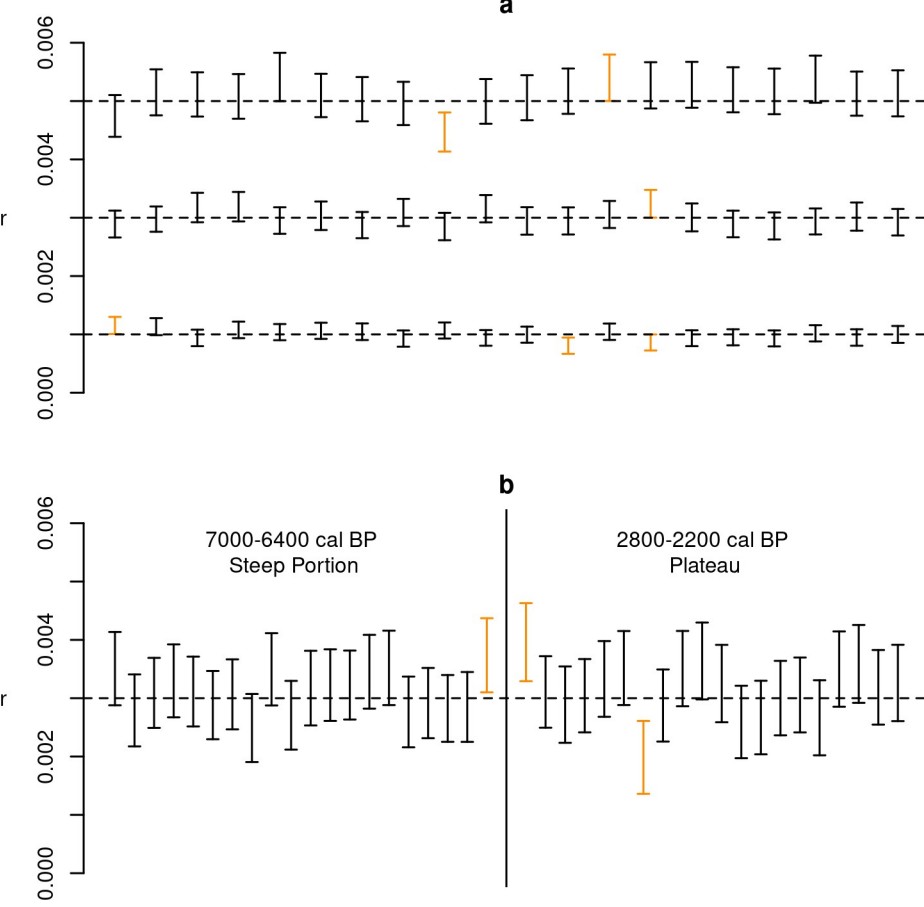

**Fig 3. Results of experiments 1 & 2.** Error bar represents the estimated 90% HPD interval for each run; cases where the true parameters were not within the interval are displayed in orange.

growth rate *r* with a median posterior of 0.0014, i.e. an annual growth rate of 0.14%. Growth estimates of $r_2$ in models **m2** and **m3**, which most likely captures the growing population trend during the Yayoi period, yielded slightly higher estimates approaching an annual growth

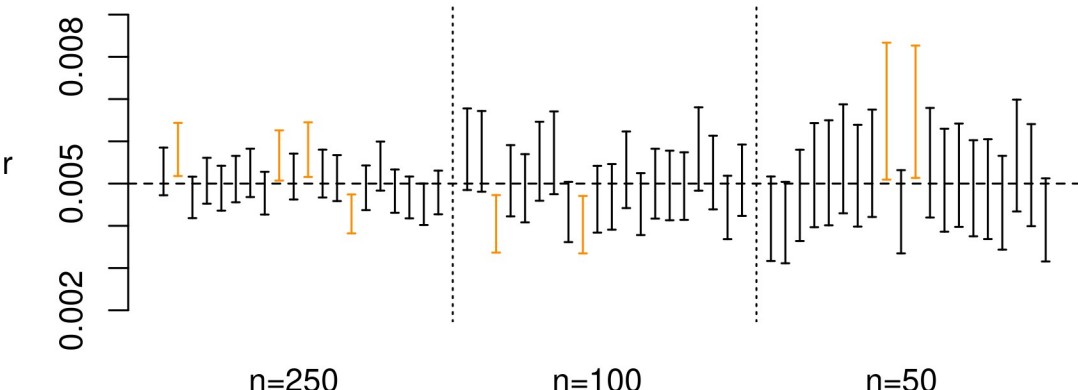

**Fig 4. Results of experiment 3a.** Error bar represents the estimated 90% HPD interval for each run; cases where the true parameters were not within the interval are displayed in orange.

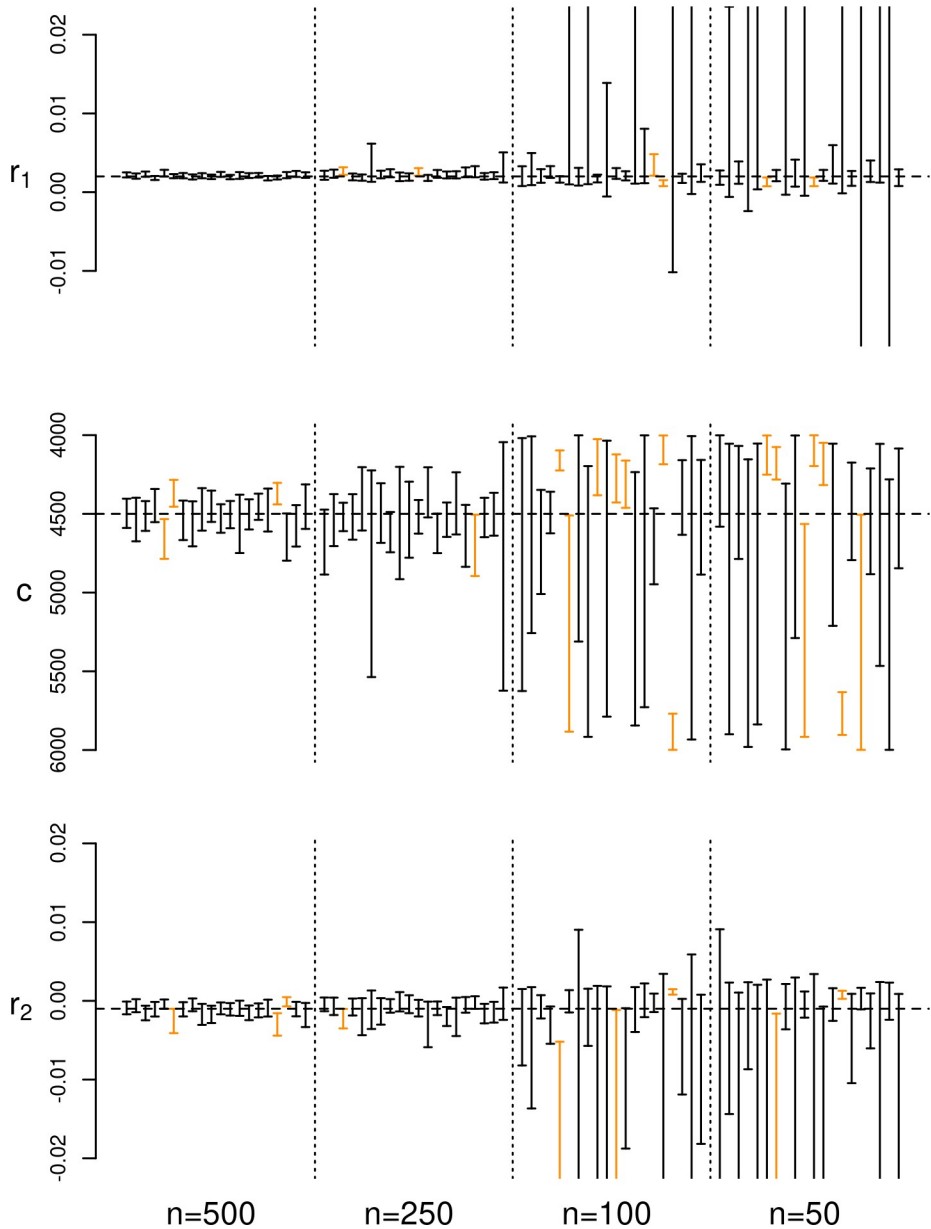

**Fig 5. Results of experiment 3b.** Error bar represents the estimated 90% HPD interval for each run; cases where the true parameters were not within the interval are displayed in orange.

rate between 0.2 and 0.3%. The smaller sample sizes associated with the first growth phase (parameter $r_1$ in **m2** and **m3**) has led to a wider 90% HPD including negative and positive rates, although in both cases, the median growth rate was negative. Both **m2** and **m3** have identified the change point $c$ in the growth rate between the end of the 8th century and the beginning of the 7th century BCE, with a similar 90% HPD interval between late 9th century and early 6th century BCE. The fitted models (Fig 8) thus indicate a growing density of radiocarbon dates over time, with both **m2** and **m3** suggesting a possible declining trend during the Final Jomon period, followed by a steep growth starting around the Early Yayoi period.

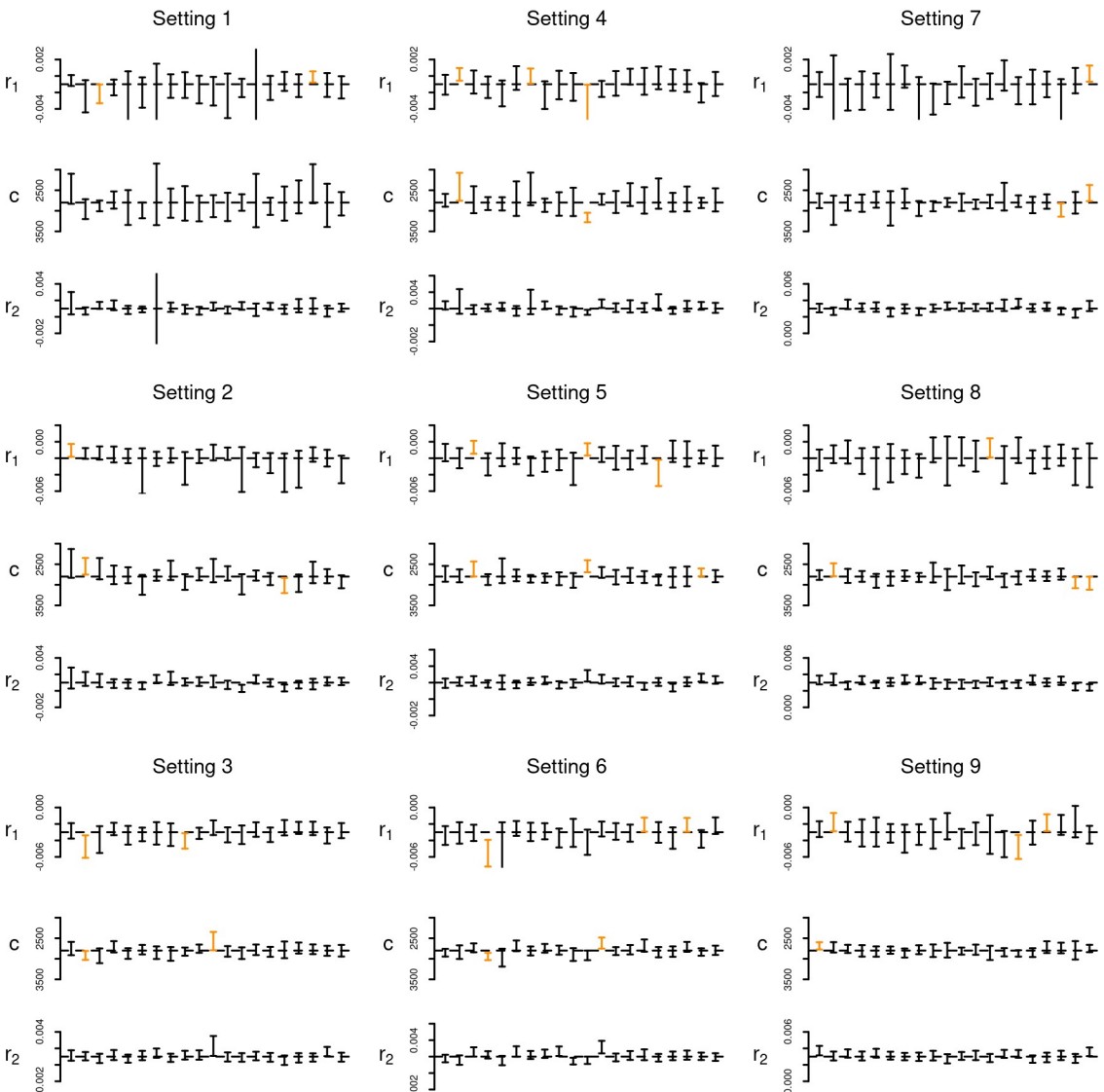

**Fig 6. Results of experiment 4.** Error bar represents the estimated 90% HPD interval for each run; cases where the true parameters were not within the interval are displayed in orange.

WAIC-based model comparison (Table 3) does not, however, suggest a single best growth model. The lack of a clear decrease in the growth rate during the logistic phase has effectively made **m2** and **m3** very similar (Fig 8), and this is reflected by similar WAIC values for the two models. Because of the small sample size, **m1** has some support (WAIC weight of 0.19), suggesting that a simple exponential growth could also explain the observed pattern, although this is less likely compared to the two growth rate alternatives portrayed by **m2** and **m3**. Posterior predictive checks of the three models (Fig 9) confirm these findings, with the observed SPD showing positive and negative deviations from the simulation envelope generated from **m1**, and a reasonably good fit for **m2** and **m3**, albeit with higher-than-expected density of radiocarbon dates around the 2nd century BCE.

**Table 2. Fitted parameter summaries.**

| Model | Parameters | Rhat | ESS | Median | HPD90lower | HPD90upper |
|---|---|---|---|---|---|---|
| m1 | $r$ | 1 | 38776.13 | 0.00137 | 0.00113 | 0.00161 |
| m2 | $r1$ | 1 | 25206.76 | -0.00021 | -0.00074 | 0.00028 |
| | $r2$ | 1 | 19585.14 | 0.00230 | 0.00184 | 0.00279 |
| | $c$ | 1 | 18148.11 | 715 | 837 | 596 |
| m3 | $r1$ | 1 | 26375.55 | -0.00029 | -0.00079 | 0.00022 |
| | $r2$ | 1 | 10178.13 | 0.00288 | 0.00196 | 0.00394 |
| | $k$ | 1 | 12127.94 | 0.07741 | 0.00012 | 0.15120 |
| | $c$ | 1.001 | 15459.57 | 696 | 812 | 582 |

The parameter $c$ is expressed in BCE.

## 5. Discussion

Recent applications of the so-called 'dates as data' approach [35] suggest a clear interest in moving beyond simple visual assessments of SPDs or basic NHST. Simple models, such as exponential or logistic growth, can be expected only under scenarios where key demographic processes and variables remain stationary over time, and an overwhelming number of studies have unsurprisingly shown that such null models can be easily rejected. Even if we assume that fluctuations in the frequency of radiocarbon dates are predominantly the result of changes in human population size, the temporal scale of the archaeological record would unquestionably cover a time window where social, economic, cultural, and environmental transformation would lead to some changes in demographic processes. From a multi-millennial perspective,

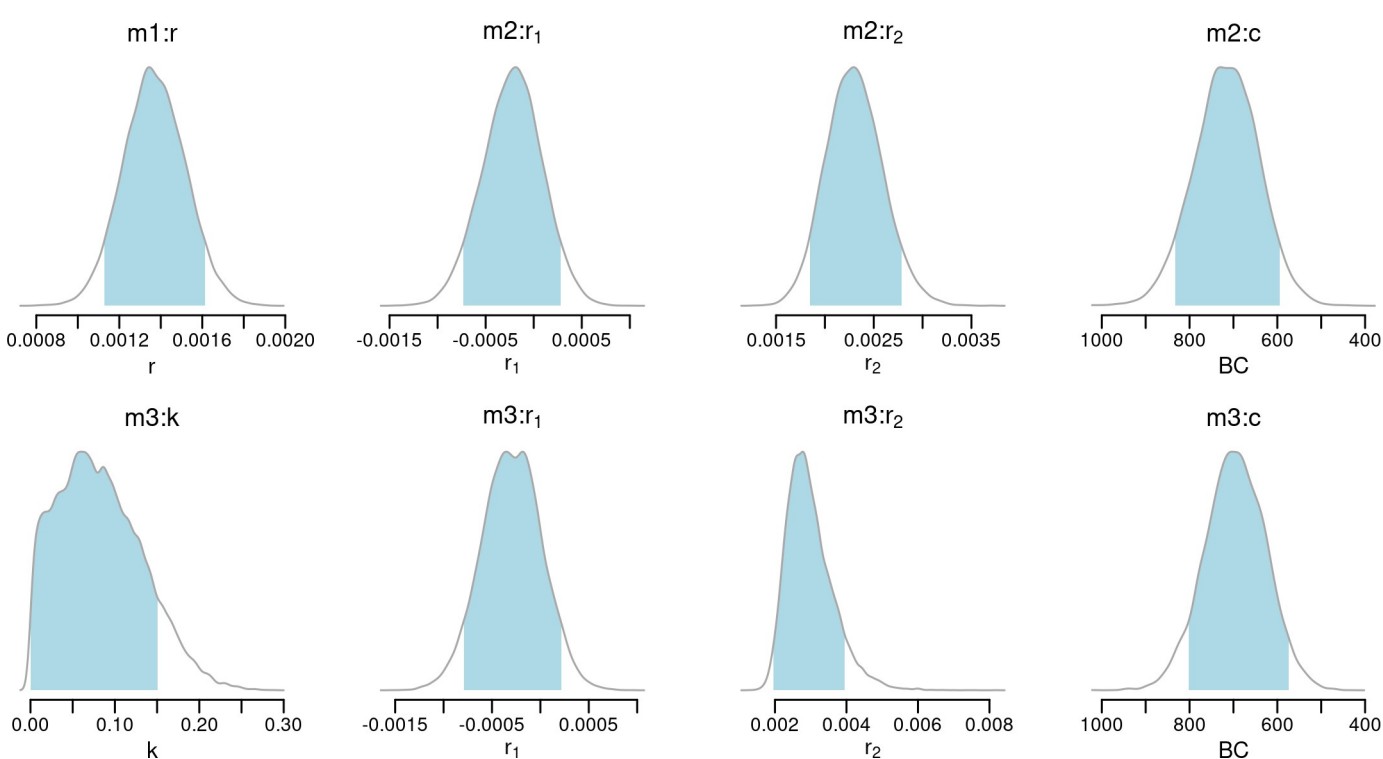

**Fig 7. Marginal posterior distributions of fitted growth parameters for models m1-m3.**

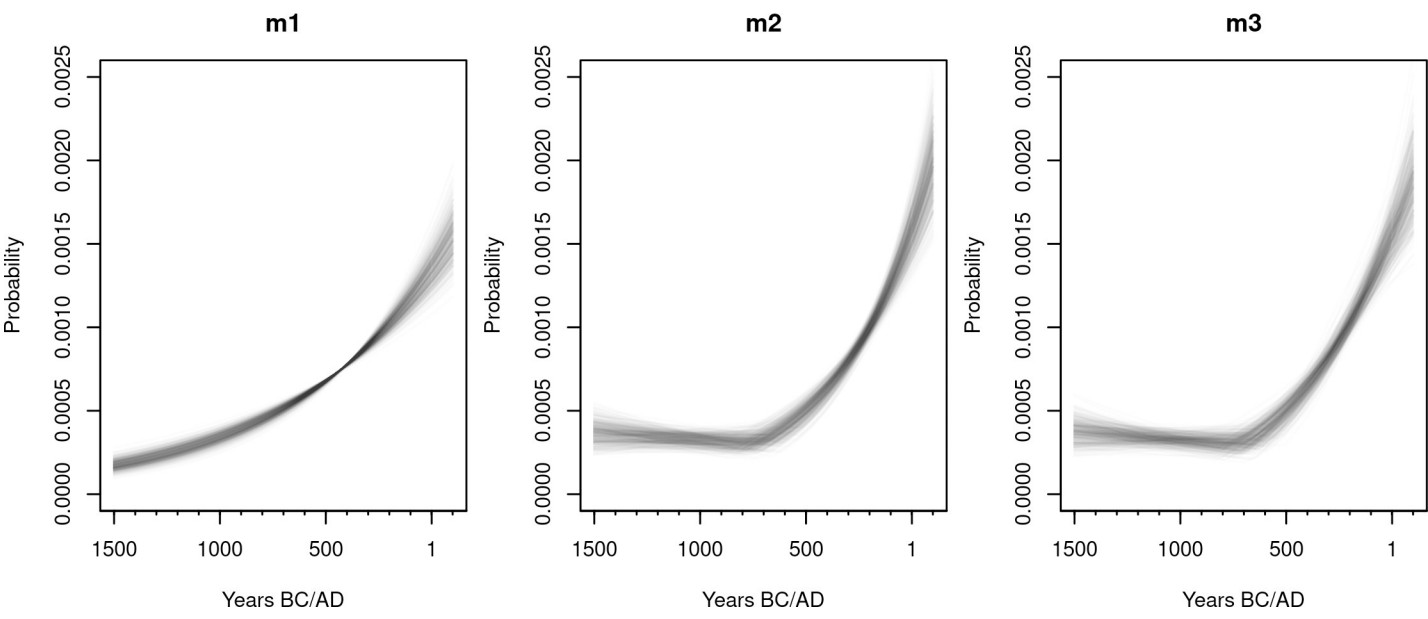

**Fig 8. Spaghetti plot with the 500 posterior samples of the fitted models.**

these changes can often be seen as relatively minor deviations from broader exponential or logistic trends. At smaller scales, however, these deviations become significant in revealing and capturing key putative processes of interest, often highlighting the importance of more nuanced statistical approaches designed to get reliable estimates of growth rates and association to external variables, as well as to compare contrasting hypotheses formally. Thus, it is not surprising to see how a number of recent studies have focused on interpreting more directly the estimated parameters [25, 76, 88] or employing methods such as information criterion to compare alternative models [10, 15, 19, 40–43]. This is undoubtedly an important step forward and has already led some authors (e.g. [15]) to explore more sophisticated models grounded on theoretical population ecology, and foreshadows an important opportunity for pursuing a model-based prehistoric demography.

Many of these applications directly utilise SPDs as a dependent variable, and as such, estimates do not account for sampling error or calibration effects, potentially leading to biased inference, particularly when dealing with smaller sample sizes and shorter time intervals. As for the Monte-Carlo NHST approaches, off-the-shelf solutions are hardly applicable in these cases, and bespoke solutions are required [10, 11, 89].

This paper is a contribution to this research agenda. It proposes a solution that addresses the issue of model fitting and model comparison within a Bayesian framework, and at the same time, makes an explicit connection with the rich literature of Bayesian phase modelling. Experimental analyses (Figs 2 to 5) have shown that our method is capable of recovering true parameters from simulated data under a variety of conditions, even in the presence of small

**Table 3. Summary of WAIC based model comparison.**

| Model | WAIC | ΔWAIC | Weights |
|-------|------|-------|---------|
| m3 | 3036.756 | 0.000 | 0.42 |
| m2 | 3036.933 | 0.177 | 0.39 |
| m1 | 3038.307 | 1.551 | 0.19 |

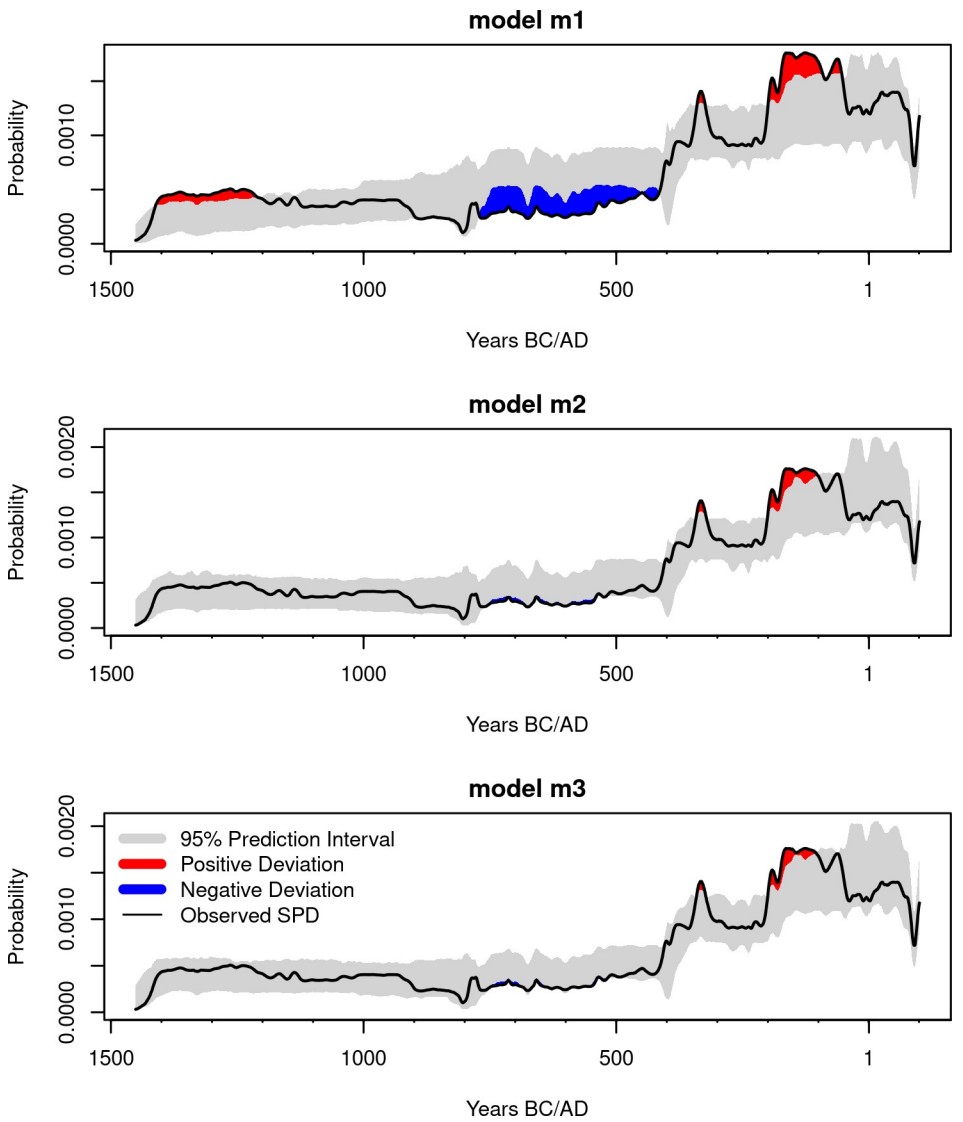

**Fig 9. Posterior predictive checks.** The 95% simulation interval of each model were obtained from SPDs using 500 parameter combinations from the posterior samples. Samples were generated using the *calsample* algorithm in *rcarbon* [8].

sample sizes and different calibration effects. Our experiments have also shown how the required sample size is a function of the nature and the complexity of the model, as well as the true target value of the parameters. For example, detecting growth rates and change-points of a double exponential model with a steep rise and fall will most likely require a smaller sample size compared to the same model with a small change in positive growth rates that resembles a standard exponential model. These findings strongly suggest that any empirical analyses should be preceded by *what-if* simulations [90] similar to what we did in experiment 4. This would allow determining the inferential power of the available sample (both in terms of size and measurement error) in relation to the information loss dictated by the calibration curve and the requirements of the proposed model.

Our method can be easily extended to other growth models drawn from the rich literature of theoretical population ecology [91], consider multiple calibration curves, and include

external covariates such as environmental data or even spatial effects. From a purely methodological point of view, there are some outstanding challenges pertaining edge effects and inter-site variation in sampling intensity. The former can be addressed by selecting samples based on their cumulative probability within the window of analyses and alternatively (or concurrently) by inferring the boundary parameters $a$ and $b$ via MCMC. The problem of inter-site variation in sampling intensity is a more complex issue. Solutions based on the binning of closely distanced radiocarbon dates [37] introduces a trade-off where one needs to sacrifice any evidence of inter-site size variation [8]. Dates referring to the exact same event can potentially be combined using the same approach as OxCal's *R_Combine* method, or alternatively by selecting one of the dates (see *nimbleCarbon*'s vignette [82] for an example). In theory, more complex models that consider the non-independent nature of the samples could be developed by using a hierarchical model where dates from the same sites are treated as a cluster.

Our case study has examined the timing and the magnitude of the population boom of the Yayoi period in the island of Kyushu. As discussed in section 1.3, previous approaches based on the temporal frequency of typologically dated settlement [65, 75] and burial [74] data, or more complex models based on putative simulated demographic processes [68, 69] have required revisions [70, 72] in response to the new estimates on Yayoi chronology, and as such, are intrinsically dependent on an indirect inferential chain connecting radiocarbon dates to the use of a particular pottery style, and recovered potsherds to specific events (e.g. the construction of a house). In contrast, our study represents the first attempt in examining the earliest population dynamics of the Yayoi period using the 'dates as data' approach, focusing specifically on the magnitude of the Yayoi population boom and the timing of its occurrence. Our growth estimates between 0.18 and 0.39% (see Table 3) per year are closely in line with the long-term estimates of 0.26% between Final Jomon and Yayoi periods estimated by Koyama [65] and smaller but on the same order of magnitude of the 0.8% estimate provided by Fujio's re-analysis of the settlement data from Mikuni Hill in northern Kyushu [72]. While discrepancies between growth rates computed from settlement data and radiocarbon dates can occur (see for example [52]), it is very likely that these differences are at least in part the consequences of spatial averaging [89], particularly given how the adoption of rice farming has occurred later in the southern parts of the island [61].

Estimates of the growth rate during the Final Jomon and Initial Yayoi period have more uncertainty, with the 90% HPD interval ranging from negative to positive values. Nonetheless, the extremely small and possible negative growth trajectories in this stage, compared to the higher rates observed in the latter half of the window of analysis, seem to support the possibility that even a small migrant population can still lead to the observed proportion of Jomon vs Migrant-type Yayoi individual recorded in the Middle Yayoi period [70], although a more detailed study on the extent of demic vs cultural diffusion and its impact of demographic processes ought to be carried out. Such follow-up studies are particularly relevant given the scarcity of human remains dated to the first half of the 1st millennium BCE, as well as their limited context of recovery (i.e. jar burials) which severely hinders a more direct assessment of the size and role played by the early migrant communities [92].

Our analyses provide some support for models suggesting the presence of a shift in growth rate occurring between the end of the 8th century and the beginning of the 7th century BCE. Comparisons between the estimated change points for models **m2** and **m3** and the earliest evidence of rice in Kyushu [93] indicate that the former occurred most likely later, with the 90% HPD of the difference between the posterior samples of the two events ranging between -15 and 241 years for model **m2** and between 14 and 256 years for model **m3** (see S7 Fig). This confirms the interpretation that that demographic boom was promoted by the spread of irrigated rice farming in the region but also suggest that this impact was not necessarily

immediate, with a possible time lag of one or two centuries. Possible reasons for this delay remain speculative at this stage as a wide range of cultural and behavioural processes could determine such temporal offset. However, it is worth noting that paleoenvironmental reconstructions suggest a temperature minimum around the mid 8[th] century BCE [94], and the subsequent warming might have supported a faster uptake of irrigated rice farming and an increase in population growth rate. Another key element that needs to be accounted for is the fact that archaeological evidence suggests multiple migratory events, the first one around 9[th]-8[th] century BCE and a second wave during the 7[th] and 6[th] century BCE [95]. If we assume these absolute chronologies to be correct, the growth rate subsequent to the change point might have been at least in part determined by migration and not just by changes in reproductive and mortality rates. Furthermore, the transition from a sporadic presence of bronze and Korean style pottery during the first few centuries of the millennium to more prominent roles in the archaeological assemblages after the 5th century BCE [96, 97], indicates the possibility of a further population influx behind the observed growth during the later stages of our window of analyses. More in general, the case study confirms the usefulness of detecting change points in the density of radiocarbon dates (see also [10, 98]) rather than exclusively focusing on growth rates, providing a framework for evaluating event-based hypotheses.

## 6. Conclusion

Statistical analyses of time frequencies of radiocarbon data have seen substantial development over the last decade with a widening of objectives and scopes in the solutions offered. Here we introduced and examined the robustness of a novel Bayesian approach that can infer parameters of population growth models, quantify their uncertainty, and provide means for pursuing formal model comparisons whilst taking into account sampling error and calibration effects. Our approach, implemented in the R package *nimbleCarbon* [82], can potentially be extended to growth models beyond those analysed in this paper and can help identify growth rates, change points, or potentially even the impact of external covariates.

Our case study has showcased how the proposed method can provide key estimates even when the number of available dates is comparatively small, allowing scholars to be stricter in their data hygiene practices. The results we obtained are in line with the archaeobotanical evidence, suggesting that there was a considerable shift in population growth rate ca 100 years after the earliest evidence of rice-farming. Due to the short time window of analyses, our population growth estimates are greater and not directly comparable to those observed for other contexts of hunter-gatherer to farming transition [76], but still offers a solid foundation for pursuing more nuanced analysis on the role of demic diffusion in the Jomon-Yayoi transition as well as the demographic impact of farming.

## Supporting information

**S1 Fig. Geographic distribution of the radiocarbon dates in Kyushu, Japan.** Base maps made with Natural Earth.
(PDF)

**S2 Fig. Double-Exponential growth models used in experiment 4.**
(PDF)

**S3 Fig. Prior predictive checks.**
(PDF)

**S4 Fig. Model m1 trace plots.**
(PDF)

**S5 Fig. Model m2 trace plots.**
(PDF)

**S6 Fig. Model m3 trace plots.**
(PDF)

**S7 Fig. Time difference between earliest dated evidence of charred rice in Kyushu [93] and inferred change-point in growth rate for models 2 and 3.** Differences were computed by comparing 15,000 random pairs of posterior samples from the two events. The highlighted region represents the 90% HPD density interval. Rice dates were calibrated using IntCal20 and combined using the R_Combine function in OxCal (via the OxCAAR R package).
(PDF)

## Acknowledgments

We are grateful to the constructive and supportive comments of the two anonymous reviewers and Erick Robinson, as well as feedbacks and suggestions on the manuscript by Robert Di Napoli and Marko Porčić. We would also like to thank Andrew Bevan and Adrian Timpson for insightful discussions concerning methodological aspects of the work.

## Author Contributions

**Conceptualization:** Enrico R. Crema.

**Formal analysis:** Enrico R. Crema.

**Funding acquisition:** Enrico R. Crema.

**Methodology:** Enrico R. Crema.

**Software:** Enrico R. Crema.

**Visualization:** Enrico R. Crema.

**Writing – original draft:** Enrico R. Crema, Shinya Shoda.

**Writing – review & editing:** Enrico R. Crema, Shinya Shoda.

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
