## [Decision Letter · Decision Letter 0]

6 Apr 2021

PONE-D-21-02107

A Bayesian approach for fitting and comparing demographic growth models of radiocarbon dates: a case study on the Jomon-Yayoi transition in Kyushu (Japan).

PLOS ONE

Dear Dr. Crema,

Thank you for submitting your manuscript to PLOS ONE. After careful consideration, we feel that it has merit but does not fully meet PLOS ONE’s publication criteria as it currently stands. Therefore, we invite you to submit a revised version of the manuscript that addresses the points raised during the review process.

Thank you for submitting your paper to Plos One. We have now received three reviews from referees with a diverse set of experiences and backgrounds. All three reviewers find that the paper essentially meets Plos One's criteria for a scientifically sound paper, and, thus, merits publication. However, the reviewers do raise several comments that may help improve the paper. Thus, I have made a minor revisions designation. Reviewer 2 raises important points about making the theory behind the models fit to the shapes of the SPDs more explicit; and Reviewer 3 raises a point worth clarifying. Again, thank you for supporting Open Science by submitting your work to Plos One.

We look forward to receiving your revised manuscript.

Kind regards,

Jacob Freeman

Academic Editor

PLOS ONE

Journal Requirements:

3. In your manuscript, please provide additional information regarding the specimens used in your study. Ensure that you have reported specimen numbers and complete repository information, including museum name and geographic location.

For more information on PLOS ONE's requirements for paleontology and archaeology research, see https://journals.plos.org/plosone/s/submission-guidelines#loc-paleontology-and-archaeology-research

4. We note that S1 Figure in your submission contains map images which may be copyrighted.

We require you to either (a) present written permission from the copyright holder to publish this figure specifically under the CC BY 4.0 license, or (b) remove the figure from your submission:

a. You may seek permission from the original copyright holder of S1 Figure to publish the content specifically under the CC BY 4.0 license. 

b. If you are unable to obtain permission from the original copyright holder to publish this figure under the CC BY 4.0 license or if the copyright holder’s requirements are incompatible with the CC BY 4.0 license, please either i) remove the figure or ii) supply a replacement figure that complies with the CC BY 4.0 license. Please check copyright information on all replacement figures and update the figure caption with source information. If applicable, please specify in the figure caption text when a figure is similar but not identical to the original image and is therefore for illustrative purposes only.

Reviewers' comments:

Reviewer's Responses to Questions

**Comments to the Author**

1. Is the manuscript technically sound, and do the data support the conclusions?

Reviewer #1: Yes

Reviewer #2: Yes

Reviewer #3: Yes

2. Has the statistical analysis been performed appropriately and rigorously? 

Reviewer #1: I Don't Know

Reviewer #2: Yes

Reviewer #3: Yes

3. Have the authors made all data underlying the findings in their manuscript fully available?

Reviewer #1: Yes

Reviewer #2: Yes

Reviewer #3: Yes

4. Is the manuscript presented in an intelligible fashion and written in standard English?

Reviewer #1: Yes

Reviewer #2: Yes

Reviewer #3: No

5. Review Comments to the Author

Reviewer #1: Population growth between the Jomon and Yayoi periods is an important question to understand the Neolithic/Bronze Age transition in Japan. This paper makes a major contribution to this issue, using new statistical methods to analyse radiocarbon dates.

Reviewer #2: Yet again, the authors have produced an important tool that advances the use of radiocarbon SPDs as a proxy measure of prehistoric demography. This is possibly their most creative and cutting edge of all their great contributions to SPD research. It certainly merits publication in a high impact journal such as PlosOne. However, while the basic math and rationale makes sense to me, I have some issues with the lack of theoretical discussion in this paper. This is reflected in their general framing of the problem they address. While they rightly frame the problem in terms of the use of SPDs, they do not discuss the actual theoretical issues involved in using an SPD as a relative, coarse-grained measure of prehistoric demography. A persistant problem for me with the Null Hypothesis Statistical Testing approach and other recent advances to SPD analysis has been the lack of consideration of these null models for what they mean in the broader context of Population Ecology Theory (cf. Turchin, Complex Population Dynamics, 2003). The exponential and logistic models have very specific general theory attached to them that the authors of this paper do not consider. Bayesian statistics are predicated on the ability to use logically-sound prior information to formalize the radiocarbon calibration process (cf. Buck and Meson, World Archaeology, 2015). There is a specific philosophy to Bayesian approaches (Buck and Meson, 2015), which the authors do not consider.

What is the basic math and theory behind an exponential growth model? Behind a Double-exponential model? I worry that, while this is another excellent technical contribution to the use of SPDs, there’s still a lack of general discussion of what these various growth models actually mean for general population ecology theory. Technical advances are important to ensure researchers do not misinterpret trends in radiocarbon SPDs. But, what about the broader theoretical implications of how these technical advances might be taken up and implemented by other researchers? I know that the authors are also skilled at theory building, which is why I challenge them to more fully articulate the growth models they employ within their new method within the broader philosophies of Bayesian statistics, and, General Population Ecology theory. This paper has the potential to be a game-changer, especially compared to the other recent advances that they mention, which in my opinion, are still vague on these general theoretical levels. New technical advances are one thing, but logical clarity in their background theory ultimately enhances their proper use by other researchers down the line. I would be very curious how Bayesian pioneers such as Caitlin Buck, or Population Ecologists such as Peter Turchin, would read this paper. In my opinion, the discussion of the different growth models used should be discussed much earlier in the paper than on page 17. And, when the authors do this, the text is very dense with math, but not accompanied by general theory. This potentially limits the impact of this paper, as many readers might not understand what’s going on. Which, increases the potential that users might mis-use the technical advances down the line. The opening part of the discussion is on-point, and articulates the current problems with NHST very well. However, this kind of discussion must be integrated in the initial framing of the problem in the introduction. It’s quite frustrating to me to read on page 24 that “Our method can be easily extended to other growth models”. What are these specific growth models? Are they based in the general theory of population ecology?

As a more specific question, I’m wondering why the r parameters used in the simulations are not similar to the prior produced rates they cite for the Jomon-Yayoi transition? I understand that these were produced using other proxies, but actually, this helps to strengthen the logic of the prior parameters built into the simulation experiments. These of course seem abnormally high to me, and the parameters the authors use in this paper are closer to realistic (although, they’re very low). But, I would be curious what would result from the authors using a wider range of parameter values for r.

Also, the authors do not explain their rationale for radiocarbon sample selection criteria excluding standard errors larger than 100 years. This seems awfully precise compared to standard practices that exlude errors <200 years or <300 years. I’m not saying the authors should not do this, but that they simply need to justify why they do.

Certain techniques are not clearly explained to the reader. An example of this is the Gelman-Rubin’s convergence diagnostic, or, the WAIC. What do these actually do? The authors mention at the top of the paper the insufficiency of AIC or BIC approaches, and propose that WAIC is better. But, there is no general discussion from the start of what that is. The authors cannot expect the readers to be as smart and savvy with statistical information criterion as they are. Different information criterion are actually very new to archaeology, and much of the field is still not using them to evaluate the performance of different models. I say all this because, as I mentioned above, this paper has the potential to be a ‘game-changer’. But, if the content is too dense and not made easily understandable to a wide-audience, then it risks: 1) not being used by as many researchers as it deserves; 2) being mis-used and mis-interpreted by researchers down the line.

On page 20 the authors note that they did not try running a larger number of MCMC iterations, but then they state that “the results do seem to suggest that an ideal minimum sample size is specific to the model and its parameters”. This is confusing to read, and raises a red-flag due to the lack of elaboration. I feel like there needs to be more sensitivity analysis in this part of the work.

Overall, this is a technically creative, savvy, and very important contribution to SPD research. I commend the authors on their continued important contributions to this field. I firmly believe that the overall contribution of this paper can be enhanced by doing what other SPD advances have not done: actually couching the rationale behind the technical advances within 1) Population Ecology theory, and 2) Bayesian philosophy.

Reviewer #3: This paper demonstrates an statistically sound investigation into the plausible shapes of the distributions used to model archaeological phases. The choices of priors distributions for parameters is well motivated, as well as the choices for hyper parameters. It is clear what packages were used within R, should someone wish to replicate the results. The reason for answering no to it being written in an intelligible fashion is simply because the sentence covering lines 479 to 480 on page 19 is confusing. Otherwise it was very well written and clear to the reader. I would be interested to see how well the methods hold up with radiocarbon dates that have a higher error, or for older parts of the calibration curve with more error. However, given the depth of what is covered already in the paper, I understand that it would be unnecessary to include that within this specific piece of research.

6. PLOS authors have the option to publish the peer review history of their article (what does this mean?). If published, this will include your full peer review and any attached files.

Reviewer #1: No

Reviewer #2: No

Reviewer #3: No

---

## [Author Response · Author response to Decision Letter 0]

14 Apr 2021

Please see the attached response to the reviewers pdf document.

---

## [Decision Letter · Decision Letter 1]

3 May 2021

A Bayesian approach for fitting and comparing demographic growth models of radiocarbon dates: a case study on the Jomon-Yayoi transition in Kyushu (Japan).

PONE-D-21-02107R1

Dear Dr. Crema,

We’re pleased to inform you that your manuscript has been judged scientifically suitable for publication and will be formally accepted for publication once it meets all outstanding technical requirements.

Kind regards,

Jacob Freeman

Academic Editor

PLOS ONE

Additional Editor Comments (optional):

Thank you for supporting open science by submitting to Plos One. It is a very nice paper that will prove useful for many researchers.

Reviewers' comments:

Reviewer's Responses to Questions

**Comments to the Author**

1. If the authors have adequately addressed your comments raised in a previous round of review and you feel that this manuscript is now acceptable for publication, you may indicate that here to bypass the “Comments to the Author” section, enter your conflict of interest statement in the “Confidential to Editor” section, and submit your "Accept" recommendation.

Reviewer #2: All comments have been addressed

Reviewer #3: All comments have been addressed

2. Is the manuscript technically sound, and do the data support the conclusions?

Reviewer #2: Yes

Reviewer #3: Yes

3. Has the statistical analysis been performed appropriately and rigorously? 

Reviewer #2: Yes

Reviewer #3: Yes

4. Have the authors made all data underlying the findings in their manuscript fully available?

Reviewer #2: Yes

Reviewer #3: Yes

5. Is the manuscript presented in an intelligible fashion and written in standard English?

Reviewer #2: Yes

Reviewer #3: Yes

6. Review Comments to the Author

Reviewer #2: Thank you for addressing the comments. This is a really nice contribution, and I commend the authors. The only remaining thing I would ask the authors build in at the proof stage is more elaboration of what the different types of WAIC results mean in Table 3 (what do the numbers in these different columns actually mean?). The authors can't assume that all their readers will be familiar with how models are compared and evaluated with WAIC.

At proof stage the authors should also check for grammatical errors, as there are some scattered throughout the text. Also check the spelling of names in acknowledgments.

Reviewer #3: (No Response)

7. PLOS authors have the option to publish the peer review history of their article (what does this mean?). If published, this will include your full peer review and any attached files.

Reviewer #2: No

Reviewer #3: No

---

## [Editor Report · Acceptance letter]

10 May 2021

PONE-D-21-02107R1 

A Bayesian approach for fitting and comparing demographic growth models of radiocarbon dates: a case study on the Jomon-Yayoi transition in Kyushu (Japan). 

Dear Dr. Crema:

I'm pleased to inform you that your manuscript has been deemed suitable for publication in PLOS ONE. Congratulations! Your manuscript is now with our production department. 

Kind regards, 

on behalf of

Dr. Jacob Freeman 

Academic Editor

PLOS ONE